# Magnesium and Pain

**DOI:** 10.3390/nu12082184

**Published:** 2020-07-23

**Authors:** Hyun-Jung Shin, Hyo-Seok Na, Sang-Hwan Do

**Affiliations:** 1Department of Anesthesiology and Pain Medicine, Seoul National University Bundang Hospital, Seongnam-si, Gyeonggi-do 13620, Korea; hjshin.anesth@gmail.com (H.-J.S.); hsknana@gmail.com (H.-S.N.); 2Department of Anesthesiology and Pain Medicine, Seoul National University College of Medicine, 103, Daehak-ro, Jongno-gu, Seoul 03080, Korea

**Keywords:** analgesia, magnesium, pain

## Abstract

In terms of antinociceptive action, the main mode of action of magnesium involves its antagonist action at the *N*-methyl-d-aspartate (NMDA) receptor, which prevents central sensitization and attenuates preexisting pain hypersensitivity. Given the pivotal function of NMDA receptors in pain transduction, magnesium has been investigated in a variety of pain conditions. The oral and parenteral administration of magnesium via the intravenous, intrathecal, or epidural route may alleviate pain and perioperative anesthetic and analgesic requirements. These beneficial effects of magnesium therapy have also been reported in patients with neuropathic pain, such as malignancy-related neurologic symptoms, diabetic neuropathy, postherpetic neuralgia, and chemotherapy-induced peripheral neuropathy. In addition, magnesium treatment is reportedly able to alleviate fibromyalgia, dysmenorrhea, headaches, and acute migraine attacks. Although magnesium plays an evolving role in pain management, better understanding of the mechanism underlying its antinociceptive action and additional clinical studies is required to clarify its role as an adjuvant analgesic.

## 1. Introduction

The first use of magnesium in medicine dates back to the 17th century [1]. Epsom salt, the major ingredient of magnesium sulfate, was used to treat conditions such as abdominal pain, constipation, and muscle strains. In modern medicine, magnesium is also widely used for the prophylaxis and treatment of pain [2].

*N*-methyl-d-aspartate (NMDA) receptors have long been the target of studies on the initiation and maintenance of central sensitization after nociceptive stimulation [3]. Magnesium and ketamine are two main NMDA receptor antagonists. Since magnesium can regulate calcium entry into cells by antagonizing NMDA receptors [4], many studies have investigated its use as an adjuvant analgesic. Recent studies proposed the use of NMDA receptor antagonists in the management of postoperative pain and a variety of acute and chronic pain conditions. The present review describes the pharmacologic basis of pain relief provided by magnesium ions, and surveys preclinical and clinical trials that investigated its antinociceptive effects.

## 2. Mechanism of Antinociceptive Action of Magnesium

Although magnesium has no direct antinociceptive effects, it inhibits calcium ions from entering cells by blocking NMDA receptors, resulting in an analgesic effect. This analgesic effect is related to the prevention of central sensitization caused by peripheral tissue injury [3]. Central sensitization occurs due to the enhancement of neuronal properties in the nociceptive pathways of the central nervous system. It is triggered by repetitive nociceptive afferent inputs and eventually manifests as a prolonged reduction in the pain threshold. Central sensitization leads to pain hypersensitivity, including wind-up or long-term pain potentiation; it causes pain even when peripheral stimuli are not intense and continues to cause pain even after the initiating stimuli have disappeared [5,6,7].

Increased intracellular calcium appears to play a major role in the initiation of central sensitization [8,9], and its buildup is related to various receptors on the postsynaptic neurons of the spinal dorsal horn, such as NMDA, α-amino-3-hydroxy-5-methyl-4-isoxazole propionate, and kainate receptors [6]. NMDA receptor activation has been demonstrated to be essential for inducing and maintaining central sensitization.

NMDA receptors are membrane ion channels expressed in the central nervous system. Each receptor has seven subunits that assemble into various combinations of tetrameric receptor complexes [10]. NMDA receptors play critical physiological roles in synaptic function including synaptic plasticity, learning, and memory [11]. NMDA receptors regulate the cellular inflows of Na^+^ and Ca^2+^ and the outflow of K^+^. This ligand-gated ion channel is non-competitively blocked in the resting state by magnesium ions and ketamine (phencyclidine site blockade), MK-801, memantine, and others [4,12]. In contrast, NMDA receptor channels are opened by membrane depolarization induced by the sustained release of glutamate and neuropeptides including substance P and calcitonin gene-related peptide [13,14].

Extracellular magnesium blocks NMDA receptors in a voltage-dependent manner [13]. Hence, it can prevent the development of central sensitization and abolish established hypersensitivity. Other competitive or noncompetitive NMDA receptor antagonists, such as D-CPP and MK801, also prevent and reverse the hyperexcitability of neurons produced by nociceptive afferent inputs [3,15].

In addition to central sensitization, calcium channels are reported therapeutic targets in neuropathic pain conditions [16]. Since magnesium is a natural calcium antagonist, the antinociceptive armamentarium of magnesium should include calcium channel blockade.

## 3. Magnesium and Perioperative Pain

Opioids have long been used to control acute postoperative and postprocedural pain. However, in a recent report that reviewed the clinical and administrative data of 135,379 adult patients who were administered opioids after hospital-based surgeries or endoscopic procedures, 10.6% of patients experienced adverse opioid-related events. These adverse events were related to poor outcomes, including an increased mortality rate, prolonged hospital stay, and higher 30-day readmission rates [17]. Dependence has grown on strong opioids to control acute and chronic pain over the past decade, influenced by the rising epidemic of prescription opioid abuse, misuse, and overdose-related deaths [18,19,20]. This phenomenon, called the “opioid crisis”, is evident in multiple countries, including Canada, Australia, the United States of America, and Europe [18,19,20]. Surgery and exposure to opioids are the identified factors contributing to persistent opioid dependence after operation and procedure [21]. Therefore, the development of procedure-specific analgesic strategies and multimodal opioid sparing techniques is important to reduce postoperative opioid dependency [22].

Anesthesiologists were unfamiliar with magnesium sulfate until recently. Magnesium is a critical participant in various physiological processes of the body. Therefore, much attention has focused on anesthesiology [23], resulting in many clinical trials [24,25,26,27,28,29,30,31,32,33,34,35], reviews, and meta-analyses [36,37,38,39,40,41,42,43,44]. In particular, the pain attenuation effect of magnesium was investigated to improve the outcomes in surgical patients. The first clinical study regarding the administration of magnesium sulfate during surgery was published in 1996 [45]; since then, numerous investigators have reported the pain attenuation effect of magnesium. Studies demonstrated that a perioperative continuous infusion [38,39] or a single bolus intravenous dose [35] of magnesium sulfate could provide effective analgesia after surgery. The usual infusion regimen consists of a loading dose of 30–50 mg/kg followed by a maintenance dose of 6–20 mg/kg/h [40].

A recent systematic review that analyzed data from 27 randomized controlled trials (RCTs) of 1504 patients (1966 through to September 2014) suggested that the systemic administration of magnesium during general anesthesia significantly attenuates postoperative pain intensity without increasing the risk of adverse events [42]. In addition, the administration of magnesium was shown to significantly reduce the use of analgesics in patients undergoing urogenital, orthopedic, and cardiovascular surgeries; improve intraoperative hemodynamics; and reduce extubation time in patients undergoing cardiovascular surgery [42]. Several systematic reviews and meta-analyses were performed to assess the benefit of magnesium on postoperative pain control in various surgeries. Chen et al. [44] investigated the effect of magnesium sulfate on analgesia after laparoscopic cholecystectomy through a meta-analysis with four RCTs of 263 patients and reported a reduction in early postoperative pain and the need for anesthetic after surgery. After analyzing 11 RCTs, Peng et al. described the perioperative systemic use of magnesium sulfate reducing the amount of analgesics and unpleasant experiences, including nausea, vomiting, and shivering [43].

Magnesium sulfate may not only decrease the amount of opioid consumption but also lessen pain intensity after surgery [46]. These effects of magnesium were demonstrated in various types of surgery. Jarahzadeh et al. reported that the intravenous use of magnesium sulfate (50 mg/kg) could provide effective analgesia and reduce requirement and adverse events of morphine after abdominal hysterectomy under general anesthesia. In other studies on obese patients who underwent open or laparoscopic sleeve gastrectomy, postoperative pain and opioid requirement were significantly lower in patients treated with magnesium sulfate [27,28,31]. A study of liver transplantation showed that the intravenous administration of magnesium sulfate could reduce the requirement of tramadol and the need for mechanical ventilation [32]. Abdelgalil et al. [34] suggested that the administration of a combined preoperative single dose of pregabalin (300 mg) and magnesium sulfate infusion (50 mg/kg) may be an effective method for enhancing postoperative analgesia and reducing total morphine use after thoracotomy.

Magnesium sulfate occasionally decreases the amount of anesthetic used during surgery. Ryu et al. [47] compared the administration of remifentanil and magnesium sulfate as agents for induced hypotension methods in middle ear surgery and observed that both drugs adequately controlled hypotension; however, patients who were administered magnesium sulfate experienced better analgesia. In addition, magnesium sulfate showed a sevoflurane sparing effect. The analgesia-enhancing effect of magnesium stabilizes the vital signs during the recovery period [47]. Magnesium sulfate may alleviate the risk of remifentanil-induced hyperalgesia after balanced anesthesia with remifentanil [48].

The benefits of magnesium can also be applied to surgical patients for postoperative pain control after spinal anesthesia. Patients who underwent total hip arthroplasty via spinal anesthesia showed lower postoperative pain intensity after systemic administration of magnesium sulfate [26]. The addition of even a small dose of magnesium sulfate to the intrathecal space with a local anesthetic prolonged the effect of spinal anesthesia and improved postoperative analgesia efficacy [49,50,51]. Magnesium sulfate administered via the intrathecal and epidural routes reduced the amount of analgesic required postoperatively [49].

Magnesium also potentiated the effects of intravenous regional anesthesia (Bier block) when combined with local anesthetics. Turan et al. [52] showed the improvement effect of magnesium sulfate on the quality of anesthesia and analgesia after its addition to lidocaine for a Bier block. The block duration was prolonged when magnesium sulfated was added.

Oral magnesium also affects pain control. The pain attenuation effect of orally-administered magnesium was observed by Jerkovic et al. [53] in patients who underwent surgical removal of their lower third molar. Patients who were administered oral magnesium before and after surgery showed lower pain intensity and degree of trismus. Postoperative sore throat (POST) is a common complication after tracheal intubation. The incidence of POST decreased with adequate anesthetic depth, smaller endotracheal tube use, minimal cuff-tracheal contact area, and proper cuff pressure ensured by an experienced anesthesiologist [54]. Borazan et al. [55] performed a randomized trial of an oral magnesium lozenge and reported that preoperative administration of a single dose of oral magnesium could reduce the incidence and attenuate the severity of POST. It was recently reported that a gargle containing magnesium sulfate effectively alleviated POST [56,57]. In addition, when magnesium is preoperatively applied via nebulizer, the incidence and severity of POST decreased [58,59].

Appropriate pain management is evidently an important aspect of perioperative anesthetic and surgical care. Acute surgical pain in the immediate postoperative period is a significant risk factor for chronic pain development and a key intervention target for reducing the risk of chronic postsurgical pain [60]. To decrease the incidence of chronic pain development, the use of aggressive multimodal treatment methods is recommended through the combination of regional anesthesia, analgesia, and other analgesic medications during the perioperative period [60]. Oh et al. [61] reported that magnesium sulfate administered perioperatively alleviated both acute and chronic postoperative pain. The rate of persistent postoperative pain at one year after total knee arthroplasty was 62% lower in patients who were administered magnesium sulfate.

## 4. Other Acute and Chronic Pain

### 4.1. Neuropathic Pain

Neuropathic pain is caused by any disease or lesion in the somatosensory system that results in the disordered transmission of sensory signals to the spinal cord or brain [62]. Magnesium has been suggested as an alternative treatment option for neuropathic pain in preclinical and clinical settings because it can block the NMDA receptor [63,64,65]. For example, when magnesium therapy was administered to patients presenting with low back pain with a neuropathic component, pain intensity reduced and the range of motion of the lumbar spine improved [66]. Neuropathic pain and functional disability following spinal cord injury can improve with magnesium treatment [67]. A case series showed that opioid-resistant cancer-related neuropathic pain is relieved by magnesium [68].

Neuropathic pain includes diabetic neuropathy, postherpetic neuralgia (PHN), cancer-related pain, trigeminal neuralgia, post-amputation pain, polyneuropathy, radiculopathy, post-stroke pain, and so on [62]. In the following section, we present a brief overview of the effects of magnesium on several types of representative neuropathic pain.

### 4.2. Diabetic Peripheral Neuropathy

Diabetic peripheral neuropathy, a complication caused by diabetes mellitus, is observed in 8–16% of diabetic patients [69]. Magnesium ions are involved in carbohydrate metabolism and insulin response, whereas magnesium deficiency is reportedly related to endocrine and metabolic disorders [70,71,72]. An inverse relationship exists between serum magnesium and fasting glucose or glycated hemoglobin levels [73]. In addition, lower serum magnesium levels are observed in patients with diabetic neuropathy [74], correlated with diabetic macro- and microvascular complications [73,75].

The therapeutic and preventive roles of magnesium against diabetic peripheral neuropathy were proven in several studies. Oral magnesium supplementation prevented allodynia, thermal hyperalgesia, and mechanical hypersensitivity in diabetic rat models by blocking NMDA receptors [69,76]. When nanoparticles including magnesium were administered to the experimental diabetic neuropathy rat model, the morphological abnormalities of dorsal root ganglion neurons and motor dysfunction improved [77,78]. In addition, low-level laser therapy resulted in good prognoses in diabetic peripheral neuropathy by increasing serum magnesium levels [79].

### 4.3. PHN

PHN occurs due to peripheral nerve damage by reactivation of the varicella zoster virus. The PHN occurs in 5–20% of patients with herpes zoster [80]. PHN is a chronic persistent pain characterized by dysesthesia, paresthesia, allodynia, and hyperalgesia. Treatment options include two categories: (1) topical therapy including lidocaine or capsaicin and (2) systemic therapy including antiepileptics, analgesics, antipsychotics, antidepressants, and magnesium sulfate [81]. In the first case, intractable PHN can be effectively treated with a transforaminal epidural injection of magnesium sulfate [82]. Magnesium sulfate was proven to be as effective as ketamine at controlling pain related to chronic PHN [83]. Magnesium sulfate is a potential and novel treatment option for the pain management of PHN; however, additional direct evidence is required for magnesium sulfate to become an optimal treatment strategy for PHN [84].

### 4.4. Chemotherapy-Induced Peripheral Neuropathy

Chemotherapy-induced peripheral neuropathy is a common dose-dependent side effect of traditional chemotherapeutic agents, including platinum agents, vinca alkaloids, taxanes, and epothilones, and recent new agents such as bortezomib and lenalidomide [85].

High doses of intravenous calcium-magnesium (Ca^2+^Mg^2+^) infusions have been thoroughly studied for the prevention of oxaliplatin-induced peripheral neuropathy. Since Gamelin et al. [86] reported its favorable protective effect against oxaliplatin-induced peripheral neuropathy, other trials have confirmed this finding [87,88], leading to the introduction of preventive Ca^2+^Mg^2+^ infusions in clinical practice for patients receiving oxaliplatin as part of chemotherapy [89]. However, conflicting results have been reported on the effectiveness of Ca^2+^Mg^2+^ infusions on oxaliplatin-induced peripheral neuropathy [90,91]. Meta-analyses and systematic reviews presented inconsistent results [92,93,94]; thus, the effect of Ca^2+^Mg^2+^ infusion on oxaliplatin-induced peripheral neuropathy remains inconclusive.

### 4.5. Fibromyalgia

Fibromyalgia is a common chronic pain syndrome without a specific etiology that causes fatigue, depression, and sleep disturbances. As its causes are unclear, multiple treatments including pharmacologic agents and non-pharmacologic therapies are used to manage widespread pain [95]. Several studies investigated the relationship between the elemental composition of the body and clinical parameters in patients with fibromyalgia [96,97,98]. They presented low magnesium levels versus normal controls [96,97]. In addition, the dietary intake of magnesium was lower in patients presenting with fibromyalgia [98]. Magnesium deficiency can increase levels of substance P, which is related to the pain intensity of fibromyalgia [99,100,101]. Thus, magnesium was suggested to be beneficial for symptom relief in patients with fibromyalgia [102,103].

### 4.6. Dysmenorrhea

Dysmenorrhea, defined as painful menstruation, can be classified as primary or secondary depending on the cause. In contrast to secondary dysmenorrhea caused by organic pelvic lesions, primary dysmenorrhea is related to abnormal uterine muscle contractions induced by prostaglandins without any recognized pathologic condition [104]. Treatment options for primary dysmenorrhea include analgesic medication, oral contraceptives, prostaglandin synthetase inhibitors, dietary changes, and other psychiatric management approaches [104].

In previous studies, decreased magnesium levels were observed in patients with dysmenorrhea [105,106]; thus, magnesium is a potential option for its prevention and treatment [107,108]. Although not clearly proven, the most suspected mechanisms in this setting are calcium channel antagonist activity or prostaglandin F2 biosynthesis inhibition [109]. The tocolytic effect of magnesium was already proven in vivo and in vitro [110,111]. However, the optimal dose of magnesium for the treatment or prevention of dysmenorrhea remains unclear.

### 4.7. Headache

Headache refers to pain occurring in the head, face, and neck regions. Tension-type headaches and migraines are the most common. Magnesium was suggested to play a pivotal role in the pathogenesis of these two types of headaches [112].

Migraine symptoms include a severe throbbing headache with nausea, vomiting, and extreme sensitivity to light or sound. Migraines are often accompanied by a warning symptom such as visual flashes of light, blind spots, a tingling sensation in the face, and trouble speaking, which are called auras. Although the pathogenesis of migraine remains to be fully elucidated, magnesium plays an essential role in migraine headaches by altering neurotransmitter secretion, synaptic transmission by cortical spreading depression, and platelet aggregation [113,114]. Accordingly, hypomagnesemia has been observed in patients with migraines [115,116]. Additionally, a low serum concentration of magnesium is an independent risk factor for migraine attacks [117]. There is considerable evidence that magnesium supplementation is useful for the management or prophylaxis of migraine headaches. Intravenous magnesium reportedly has an effect that is similar to or better than caffeine for treating acute migraines [118]. After magnesium medication, the incidence of migraine attacks decreased significantly [119,120,121], although there was a contrary result where magnesium had no effect on migraines [122]. These inconsistent results may be caused by differences in magnesium formulation, dose, or migraine subtype. Thus, further trials are required to recommend proper magnesium supplementation formulation or dose for each migraine type.

Tension-type headaches are usually bilateral and diffuse throughout the head, forehead, and neck. Their exact causes are not well understood; thus, it is often difficult to treat them effectively. In addition to bad posture or muscle tension, magnesium is suggested in the etiology and treatment of tension-type headaches [123]. Ionized magnesium levels of the serum, salivary secretions, and platelets are known to decrease in patients with tension-type headaches [124,125]. Magnesium treatment reportedly improved the symptoms of episodic or chronic tension-type headaches for at least one year [126,127].

## 5. Conclusions

Evidence of the role of magnesium in analgesic adjuvants against a variety of acute and chronic pain has accumulated over decades. The mechanism of the antinociceptive effect of magnesium is mainly explained by its inhibitory action on NMDA receptors and central sensitization. In addition to the direct action of magnesium on analgesia, attention should be paid to its indirect actions on the disease.

As an essential mineral nutrient, increased magnesium intake or supplementation can improve the course of some disease conditions such as osteoarthritis [128], neurological disorders [129] and cardiovascular disease [130], leading to improved analgesia. It is conceivable that this role of magnesium is much more important than improved analgesia because magnesium can be helpful for disease prophylaxis and treatment. Magnesium injections and preparations will be used more frequently in everyday clinical practice as more consistent and convincing evidence accumulates.

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
