# Peer review of "Magnesium and Pain"

_nutrients, 2020, doi:10.3390/nu12082184_

Round 1
Reviewer 1 Report
The present review addresses an important topic, as adequate control of chronic pain is not often achieved. The role of Magnesium as adjuvant analgesic is supported by literature, however the present review, with the exception of few recent cited studies, looks like a repetition of what already published (PMID: 29920000) Furthermore, there are a number of sentences in the introduction that need to be deeply revised. Below some of them. Lines 55-57: “… and the build-up of intracellular calcium is associated with various receptors on 56 the postsynaptic neurons of the spinal dorsal horn, such as NMDA, 57 α-amino-3-hydroxy-5-methyl-4-isoxazole propionate (AMPA), kainate, and glutamate receptors [6]” This sentence does not make sense because NMDA, AMPA and Kainate receptor ARE glutamate receptors. Lines 59-60: “In this regard, calcium channel blockers have shown antinociceptive effects in chronic pain conditions [10]” This reference is not appropriate since it refers to voltage-gated calcium channels, while NMDA receptors are inotropic receptors (ligand-gated ion channels). Lines 64-66: “This voltage-dependent ion channel is blocked non-competitively in the resting state by the magnesium ion and by ketamine (phencyclidine site blockade), MK-801, memantine, and others [4, 12]” I believe there is little misunderstanding about NMDA receptor that are inotropic receptors. Ionotropic receptors are different from voltage-dependent ion channels. Lines 62-63: “NMDA receptors are membrane ion channels expressed in the CNS. Each receptor is a tetramer composed of four different subunits—two NR1 and two NR2 [11]" Literature about NMDA receptors is quite obsolete as well as the nomenclature of NMDA subunits. More recent literature should be cited. For example PMID: 30037851; 28986865; 32197322
Author Response
Manuscript number: Nutrients-863129
Dear Reviewers,
Thank you very much for your comments on our article. We agree to the points reviewer indicated. The paper has been substantially revised based on the reviewer’s comments.
We hope the revised manuscript better meets the requirements of ‘Nutrients’.
We also want to thank the reviewers for their efforts in improving our paper. Our point-to-point responses to your and the reviewer’s comments are as follows
****************************************************************
EDITOR'S COMMENTS:
Reviewer #1:
Lines 55-57: “… and the build-up of intracellular calcium is associated with various receptors on 56 the postsynaptic neurons of the spinal dorsal horn, such as NMDA, 57 α-amino-3-hydroxy-5-methyl-4-isoxazole propionate (AMPA), kainate, and glutamate receptors [6]” This sentence does not make sense because NMDA, AMPA and Kainate receptor ARE glutamate receptors.
- Thank you for your comment. According to your comment the sentence has been revised.
[Lines 53-55] … and its buildup is related to various receptors on the postsynaptic neurons of the spinal dorsal horn, such as NMDA, α-amino-3-hydroxy-5-methyl-4-isoxazole propionate, and kainate receptors [6].
Lines 59-60: “In this regard, calcium channel blockers have shown antinociceptive effects in chronic pain conditions [10]” This reference is not appropriate since it refers to voltage-gated calcium channels, while NMDA receptors are inotropic receptors (ligand-gated ion channels).
- In order to make the meaning clearer, the description was changed as follows and the reference was changed.
[Lines 70-71] … In addition to central sensitization, calcium channels are reported therapeutic targets in neuropathic pain conditions [16].
Lines 64-66: “This voltage-dependent ion channel is blocked non-competitively in the resting state by the magnesium ion and by ketamine (phencyclidine site blockade), MK-801, memantine, and others [4, 12]” I believe there is little misunderstanding about NMDA receptor that are inotropic receptors. Ionotropic receptors are different from voltage-dependent ion channels.
- Thank you for your valuable comments. The above sentence was changed as follows:
[Lines 60-61] This ligand-gated ion channel is blocked non-competitively in the resting state by magnesium ions and ketamine (phencyclidine site blockade), MK-801, memantine, and others [4, 12].
Lines 62-63: “NMDA receptors are membrane ion channels expressed in the CNS. Each receptor is a tetramer composed of four different subunits—two NR1 and two NR2 [11]" Literature about NMDA receptors is quite obsolete as well as the nomenclature of NMDA subunits. More recent literature should be cited. For example PMID: 30037851; 28986865; 32197322.
- According to reviewer’s valuable comments, the above sentences were changed as follows and the references were replaced with new ones you recommended.
[Lines 57-60] NMDA receptors are membrane ion channels expressed in the central nervous system. Each receptor has seven subunits that assemble into various combinations of tetrameric receptor complexes [10]. NMDA receptors play critical physiological roles in synaptic function including synaptic plasticity, learning and memory [11].
Reviewer 2 Report
This is a well-organized and written manuscript where the authors describe the role and effect of magnesium in pain management.
As the titel is magnesium and pain without differentiating between i.v. magnesium and oral magnesium supplementation, the manuscript is very focused on the effect of magnesium sulfate or magnesium i.v.. Oral magnesium treatment is included but to a minor part. This could be improved as there are also more data on oral magnesium and pain/analgesic effect (e.g. sore throat and other indication).
Concerning the references, there is a larger number of reviews and meta-analysis cited instead of original literatur which is more unusual as this manuscript is a review itself. Especially in the section "migraine", original studies should be implicated.
Author Response
Manuscript number: Nutrients-863129
Dear Reviewers,
Thank you very much for your comments on our article. We agree to the points reviewer indicated. The paper has been substantially revised based on the reviewer’s comments.
We hope the revised manuscript better meets the requirements of ‘Nutrients’.
We also want to thank the reviewers for their efforts in improving our paper. Our point-to-point responses to your and the reviewer’s comments are as follows
***************************************************************************
EDITOR'S COMMENTS:
Reviewer #2:
This is a well-organized and written manuscript where the authors describe the role and effect of magnesium in pain management.
As the titel is magnesium and pain without differentiating between i.v. magnesium and oral magnesium supplementation, the manuscript is very focused on the effect of magnesium sulfate or magnesium i.v.. Oral magnesium treatment is included but to a minor part. This could be improved as there are also more data on oral magnesium and pain/analgesic effect (e.g. sore throat and other indication).
- Thank you for your valuable comments. We have added a paragraph regarding the effect of oral magnesium on postoperative pain and sore throat:
[Lines 142-153] Oral magnesium also has a pain control effect. The pain attenuation effect of orally administered magnesium was observed by Jerkovic et al. [53] in patients who underwent surgical removal of the lower third molar. Patients who were administered oral magnesium before and after surgery showed lower pain intensity and trismus degree. Postoperative sore throat (POST) is a common complication after tracheal intubation. The incidence of POST decreased with adequate anesthetic depth, smaller endotracheal tube use, minimal cuff-tracheal contact area, and proper cuff pressure ensured by an experienced anesthesiologist [54]. Borazan et al. [55] performed a randomized trial of the oral magnesium lozenge and reported that preoperative administration of a single dose of oral magnesium can reduce the incidence and attenuate the severity of POST. It was recently reported that a gargle containing magnesium sulfate effectively alleviated POST [56, 57]. In addition, when magnesium is applied via nebulizer preoperatively, the incidence and severity of POST decreased [58, 59].
Concerning the references, there is a larger number of reviews and meta-analysis cited instead of original literatur which is more unusual as this manuscript is a review itself. Especially in the section "migraine", original studies should be implicated.
- We agree with your opinion. The “Migraine” section has been revised.
[Lines 253-260] There is considerable evidence that magnesium supplementation is useful for the management or prophylaxis for migraine headache. Intravenous magnesium reportedly has an effect that is similar to or better than caffeine for treating acute migraine [118]. After magnesium medication, the incidence of migraine attacks decreased significantly [119-121]. However, there was a contrary result that magnesium had no effect on migraine [122]. These inconsistent results may be caused by differences in magnesium formulation or dose or migraine subtype. Thus, further trials are required to recommend proper magnesium supplementation formulation or dose for each migraine type.
Round 2
Reviewer 1 Report
The manuscript has been improved.